# The Analysis of Blood Inflammation Markers as Prognostic Factors in Parkinson’s Disease

**DOI:** 10.3390/healthcare10122578

**Published:** 2022-12-19

**Authors:** Iulia-Diana Stanca, Oana Criciotoiu, Simona-Daniela Neamtu, Ramona-Constantina Vasile, Nicoleta-Madalina Berceanu-Bora, Teodora-Nicoleta Minca, Ionica Pirici, Gabriela-Camelia Rosu, Simona Bondari

**Affiliations:** 1Department of Neurology, University of Medicine and Pharmacy of Craiova, Petru Rares 2, 200349 Craiova, Romania; 2Department of Hematology, Faculty of Pharmacy, University of Medicine and Pharmacy of Craiova, Petru Rares 2, 200349 Craiova, Romania; 3Department of Epidemiology, University of Medicine and Pharmacy of Craiova, Petru Rares 2, 200349 Craiova, Romania; 4Department of Neurology, Clinical Hospital of Neuropsychiatry Craiova, Calea Bucuresti 99, 200473 Craiova, Romania; 5Department of Anatomy, University of Medicine and Pharmacy of Craiova, Petru Rares 2, 200349 Craiova, Romania; 6Department of Research Methodology, University of Medicine and Pharmacy of Craiova, Petru Rares 2, 200349 Craiova, Romania; 7Department of Radiology, University of Medicine and Pharmacy of Craiova, Petru Rares 2, 200349 Craiova, Romania

**Keywords:** Parkinson’s disease, neuroinflammation, peripheral immune system, neutrophile-to-lymphocyte ratio

## Abstract

Parkinson’s disease is a chronic, progressive, and neurodegenerative disease, and yet with an imprecise etiopathogenesis. Although neuroinflammation was initially thought to be a secondary condition, it is now believed that microglia-induced inflammation could also contribute to the degeneration of the nigrostriatal pathway. Here, we aimed to establish the feasibility of basic inflammatory biomarkers as prognostic factors in PD. The study was based on retrospective analyses of blood samples taken from patients diagnosed with PD, as well as from healthy subjects. Complete medical records, total leukocyte count with subpopulations, and erythrocyte sedimentation rate (ESR) were analyzed. We calculated the serum neutrophils-to-lymphocytes ratio (NLR) and platelet-to lymphocytes ratio (PLR), and also compared the laboratory data between the PD group and the control group. Only PLR and NLR showed statistically significant differences (*p* < 0.001 and 0.04, respectively). In our study, ESR did not show statistically significant correlations with motor score or with disability. In our research, ESR was correlated with the disease duration (*p* = 0.04), and PLR showed a significant correlation with disease stage (*p* = 0.027) and disease duration (*p* = 0.001), but not with motor state. These biomarkers could prove to be effective tools for a primary evaluation of inflammation in PD, but further tests are required to properly investigate the neuroinflammatory status of these patients.

## 1. Introduction

Idiopathic Parkinson’s disease (PD) is a progressive, heterogeneous, and multisystem neurodegenerative disease, whose global prevalence rate increases with age [1,2,3,4]. The clinical features represented by motor symptoms include bradykinesia, rigidity, tremor, and later on postural instability, but there are also a wide array of non-motor symptoms [5].

Numerous studies have demonstrated the existence of increased levels of mono and oligomeric species of alpha-synuclein, which are involved in the pathogenesis of PD [6]. When PD progresses, due to oxidative stress and inflammation, this leads to interdependent pathological conditions involving humoral and cellular immunity, which will lead to a protein misfolding cascade [7]. Although the etiopathogenesis of the disease remains unclear, many studies point to neuroinflammation as a contributor to the development and progression of PD [8,9]. It is conventionally established that the underlying mechanism involved in the pathogenesis of PD is the degeneration of the nigrostriatal pathway, but it is unclear what causes the degeneration in this particular area. Several studies have highlighted that an increase in inflammation could have a role in the pathogenesis of this disease, but it remains to be determined whether chronic inflammation is the cause or the effect of neurodegeneration [10,11]. Chronic neuroinflammation can lead to the alteration of the blood–brain barrier, which favors the infiltration of the central nervous system (CNS) with chemokines and cells from the peripheral immune system. These would activate glial cells, T lymphocytes, and mast cells within the CNS, determining an increase in neuroinflammation, which thus becomes chronic and leads to neuronal loss. The release of neurotoxic molecules due to the activation of inflammation in the CNS and from peripheral immune cells are factors that accelerate the neurodegeneration process [12]. Proinflammatory cytokines and chemokines cause oxidative stress and damage to the dopaminergic neurons [13].

Several neuroimaging methods have proven their usefulness in evaluating neuroinflammatory activity in vivo, using specific radiotracers, e.g., [F]-FEPPA PET [14] or 11C- PK11195 PET [15]. However, as these methods cannot be used in current clinical practice, a putative solution can be sought, by evaluating the link between neuroinflammation and peripheral inflammation [16].

The serum neutrophils to lymphocytes ratio (NLR) is an inexpensive and easy to evaluate marker for peripheral inflammation. There are numerous studies that have shown that the number of neutrophils, lymphocytes, and especially their ratios are practical methods to determine the level of systemic inflammation, being used as a predictive factor for the prognosis of cardiovascular diseases, cerebrovascular diseases, neurodegenerative diseases such as Alzheimer’s disease, and amyotrophic lateral sclerosis [17,18]. C-Reactive protein (CRP) and erythrocyte sedimentation rate (ESR) are other facile methods for assessing the level of systemic inflammation. Additionally, among other methods, platelet-to lymphocytes ratio (PLR) could also be taken in consideration as an instrument used to indicate inflammation.

Altogether, we aimed to evaluate the role of the systemic inflammation status by analyzing NLR, PLR, and ESR in patients with PD, to compare them with healthy controls, and to investigate if there was an association between the clinical features of PD patients and NLR, PLR, and ESR. We thus wanted to establish the feasibility of such biomarkers as prognostic factors in PD.

## 2. Materials and Methods

Our study was based on retrospective analyses of blood samples taken from patients diagnosed with PD and from healthy subjects.

We enrolled 45 patients diagnosed with PD according to current criteria [19] and hospitalized in the Neurology Department of the Neuropsychiatry Clinical Hospital of Craiova between 1 January 2017 and 31 September 2022 (PD group). This group was composed of 19 women and 26 men aged between 46 and 80 years and with a mean age of 65.91 ± 8.65 years. In addition, 46 age- and gender-matched healthy subjects, unrelated to the families of the PD patients, from the same geographic area were chosen as the control group (CG). The demographic data, such as age, gender, height, and weight were recorded.

Depending on the severity of the disease, evaluated using the Hoehn and Yahr (HY) scale [20], the patients were divided into early-stage PD (stages 1–2,5) and advanced-stage PD (stages 3–5). To assess the motor function of these patients, we used the MDS-Unified Parkinson’s Disease Rating Scale (MDS-UPDRS) part III [21]. Patients with chronic disease, abnormal brain CTs, history of infectious diseases, or those who used medication that could have interfered with the laboratory results were excluded from the study.

All patients were assessed using the mini-mental state examination (MMSE) and geriatric depression (GDS) scales; patients with MMSE ≤24 and GDS (short form) ≥5 were excluded from the study.

Blood samples from the PD group were obtained from their medical records, and all the patients in this group were hospitalized for follow-up. In the control group, blood samples were obtained from outpatients attending our hospital for routine laboratory tests. Samples were taken after 12 h of fasting, between 8^00^ and 12^00^ in the morning. Written informed consent was obtained from all study participants before the blood samples were taken, and the Ethics Council of The Clinical Hospital of Neuropsychiatry Craiova, Romania (EC 3/22) approved the study.

Complete medical records and total leukocyte count with subpopulations (neutrophils, lymphocytes, monocytes, eosinophils, and basophils) measured in peripheral blood were analyzed. NLR was calculated as the ratio between absolute neutrophil count and absolute lymphocyte count. PLR was calculated as the ratio between absolute platelet count and absolute lymphocyte count.

### Statistical Analyses

All statistical analyses were conducted using IBM SPSS Statistics (version 20.0) and descriptive and exploratory data analyses were performed. Continuous data were first explored for normality utilizing Kolmogorov–Smirnov testing, and if the distribution was normal an independent sample *t*-test was utilized to explore the differences between the two groups; for non-normal distributed datasets, a Mann–Whitney U was used instead. Multivariable regression analyses using “age” as the main confounder were used to test if the different serum marker levels significantly predicted patients’ pathological features. The differences between the two groups were analyzed through an independent sample *t*-test. A Pearson r test was used for the correlation of the normal distribution variables and a Spearman test was used for variables without normal distribution. Through this analysis, the values of *p* < 0.05 were accepted as statistically significant.

## 3. Results

### 3.1. Comparations of Demographic Features and Laboratory Indicators of the Two Groups

#### 3.1.1. Demographic Features of the Two Groups

The PD group (n = 45) was composed of 19 women (42.22%) and 26 men (57.77%) with a mean age of 65.91 ± 8.65 years. In the control group (CG), from 46 healthy subjects, 23 were women (50%) and 23 were men (50%), and the mean age was 62.65 ± 13.38 years.

A Kolmogorov–Smirnov test indicated that all the continuous data retrieved for the patients’ group followed a normal distribution (*p* ≥ 0.068), except the ESR [D(46) = 1.768, *p* = 0.004]. For the control group, except the MMSE [D(46) = 1.715, *p* = 0.006] and ESR [D(46) = 1.835, *p* = 0.002], all the other data also followed a normal distribution (*p* ≥ 0.112) (see Table 1 and Table 2).

The demographic and clinical features of the participants are presented in Table 1. There were no obvious differences concerning the age, gender, or provenience between the two groups (*p* > 0.05). These findings indicated their comparability. However, we found a high statistically significant difference regarding the score obtained on the MMSE scale between the two groups.

Categorical variables are indicated in the table as number (percentage), whereas the continuous variables are indicated as mean ± SD.

#### 3.1.2. Laboratory Findings

We used independent sample *t* tests or Mann–Whitney U statistics to see if there was a difference between the PD patients and control group.

The WBC count, neutrophils, and platelets were not significantly different between the two groups. However, the lymphocyte counts were higher in the CG compared with the PD group, with the average NLR being significantly different between the two groups (*p* = 0.04). 

On the other hand, the PLR difference between the CG and PD groups was highly statistically significant (*p* < 0.001), but unfortunately, ESR did not show any statistical difference between the two groups (*p* = 0.187). Laboratory data are summarized in Table 2.

Regarding the PD group, we noticed that in the subgroup of patients with advanced PD, the ESR and PLR had higher values compared to early-stage PD (*p* < 0.001 * and *p* < 0.001, respectively). The NLR values did not show significant differences between the two groups, although for the advanced PD patients, this ratio had a tendency for higher values (*p* = 0.06).

#### 3.1.3. Correlations and Predictors Analysis

In the PD group, the median ESR value was positively correlated with the age of the patients (*p* = 0.006), disease duration (*p* < 0.001), NMSQ (*p* = 0.005), MDS-UPDRS score (*p* = 0.011), and severity of the disease (*p* < 0.001). That means that the inflammation measured through the ESR increased with age, disease duration, non-motor symptoms, MDS-UPDRS score, and disease severity. However, when we used multiple regression analysis, we noticed that both the ESR and NLR did not show a predictor effect, either on the score obtained on the MDS-UPDRS or on disease severity.

Multiple regression analysis was used to test if the different serum marker levels significantly predicted the PD patients’ clinical data, considering age as a potential co-predictor (Table 3).

Thus, our analysis showed that the ESR, NLR, and PLR were not significant predictors for the MDS-UPDRS score. NLR had a detectable, albeit non-significant, influence on the MDS-UPDRS score (β = 4.884, *p* = 0.062), and age alone had a significant effect on the value of the MDS score (β = 0.673, *p* = 0.119). For the HY score, both the ESR and NLR did not show a predictor effect, but there was a strong influence of the age factor over this parameter (β = 0.037, *p* = 0.046; β = 0.047, *p* = 0.008). PLR, on the other hand, showed a strong correlation with the HY score, although with a moderate amplitude (β = 0.007, *p* = 0.027), and here age did not have a significant influence over the variation of this parameter (β = 0.031, *p* = 0.088). Finally, ESR, NLR, and PLR could not predict the NMSQ variations, and age did not exhibit any significant influence on this score.

We next thought to explore the influence of age on the other different parameters in the PD group and to check if age alone had an influence on them in both the PD and the control groups (Table 4). 

Thus, age had a significant effect on the ESR value in the PD group (β = 0.655, *p* = 0.01), but not in the control group (β = −0.029, *p* = 0.451), reflecting the influence and connection between the disease pathogenesis inflammatory burden and ESR. The NL ratio did not seem to be influenced by age for both patient groups; while as for the ESR, the PLR values were significantly predicted by age in the PD group (β = 2.171, *p* = 0.008), but with no significant age influence in the control group (β = −0.007, *p* = 0.412). Age showed a negative correlation with the MMSE score, albeit non-significant, for the PD group (β = −0.038, *p* = 0.312); however, this relationship was significant for the control group (β = −0.027, *p* = 0.001). Disease duration was predicted by the PL ratio (β = 0.057, *p* = 0.001), but not by ES and NL ratios. In addition, the onset age was not a significant predictor for the UPDRS motor score and NMSQ.

## 4. Discussion

We investigated the relationship between NLR, PLR, ESR, and PD duration and stage. Many studies have shown that neuroinflammation is common in various neurodegenerative diseases, it being widely known that the microglia contribute to the secretion of reactive oxygen species (ROS), causing oxidative stress and persistent inflammatory responses [22,23]. The formation of Louis nucleosomes, mainly composed of -synuclein (-syn), is a characteristic pathological feature of PD. The increase level of α-syn in the serum is considered an effective biomarker for the diagnosis, differential diagnosis, and prognosis of PD and has an essential role, together with the microglia, in the activation of the inflammatory process [24].

Previous research has shown that neutrophils have the ability to cross the blood–brain barrier and to recruit, activate, and regulate the transport of different leukocyte populations in tissues and initiate the inflammatory response through regulation of chemokines [25]. These studies highlighted the fact that there is a statistically significant higher number of neutrophils and a considerably lower number of lymphocytes in patients with PD compared to healthy subjects, and this is partially consistent with the data we obtained. By comparing the PD group with healthy subjects, our study pointed to a higher neutrophil count in the PD group, but without attaining a significant difference. In contrast, the lymphocytes count was statistically significant lower in PD patients when compared with CG (*p* = 0.04), having as a consequence the fact that the NLR was significantly higher in patients with PD (*p* = 0.04). This was due, most probably, to the fact that in our retrospective study, we could not analyze genetic features of the patients or other markers of the inflammatory process, which could have been determined if more blood samples had been available. The lower number of lymphocytes could be explained by the fact that lymphocytes could be recruited in the brain parenchyma due to the dysfunction of the blood–brain barrier [26].

Total blood count and ESR are simple and cost-effective tests that provide significant information related to the inflammatory status within the body. Compared to other inflammatory cytokines (such as IL-6, IL-1β, and TNF-α), PLR, NLR, and ESR proved to be more easily assessable inflammatory factors for a series of chronic neurological diseases [27].

NLR has been frequently used as an inflammatory marker and prognostic indicator in a variety of neurological diseases, but its value in PD patients has given inconsistent results. In the study conducted by Akil et al., the authors showed that NLR levels were statistically significantly higher in PD patients compared to healthy subjects [27]. In our study, there were slightly statistically relevant differences between the group of patients with PD and the healthy subjects (*p* = 0.04), and this was in accordance with the study mentioned above [27], although another study did not show differences between PD and healthy subjects [28]. Our results showed that PD patients had a lower absolute number of lymphocytes and a trend towards a higher neutrophil count, compared to CG. This is in accordance with the data reported in the literature [29,30]. Considering previous studies, this pattern could indicate a loss of the immune protective function of lymphocytes in PD [30,31]. As neutrophils have been shown to be involved in chronic inflammation, in our study there was a trend of increased absolute neutrophil counts in PD patients, but without statistically significance, which is consistent with other studies [27,28,32].

In the group of patients with PD, we did not find a statistically significant dependence between NLR and the severity of the disease (evaluated by the HY scale) or disease duration. Moreover, we found no association between NLR and the MDS-UPDRS score or the motor subtypes of PD. These findings are in accordance with the previous study conducted by Muñoz-Delgado et al. [33].

In addition, our study highlighted another important aspect, that there was a statistically significant positive correlation between PLR, as an indicator of inflammation, and the disease duration (*p* = 0.001), as well as the HY stage (*p* = 0.027). According to our results, PLR is probably an effective indicator of overall peripheral immune dysregulation and inflammatory status in PD. However, we cannot pinpoint whether the alteration of PLR values is the cause or the consequence of the disease progression.

In the recent literature, the role of platelets in neurodegeneration has been widely discussed [34]. It is well known that platelets are present in the cytoplasm or in exosome neurotransmitters such as γ-amino-butyric acid (GABA), glutamate, serotonin, epinephrine, dopamine, and histamine; and this fact can substantiate the idea that these cells act as messengers that connect the CNS with the peripheral environment. 

We know that platelets release serotonin when they are exposed to specific glycolipid structures of neurons and astrocytes, and this phenomenon occurs in response to the destruction of the blood–brain barrier and leads to neuroinflammation in neurodegeneration [35]. In our study, PLR was positively correlated with disease duration (*p* = 0.001) and severity of the disease (*p* = 0.027), leading to the conclusion that inflammation increases with disease duration and as patients become more disabled. 

Due to the fact that the inflammatory processes that occurs in the pathogenesis of PD are complex and include multiple variables, it is very difficult to determine which components of the immune system play a specific role.

However, the increase of such biomarkers being utilized in everyday clinical practice as non-invasive parameters might help reflect the level of inflammation in PD.

The limitations of our study were that we only had retrospective datasets and that the PD patients and the CG were not assessed in terms of nutritional characteristics, not even from the point of view of the possible effects of medications used; and, last but not least, the relatively small size of the dataset. In addition, we suggest that further prospective studies with a larger number of patients are needed to establish the role of the peripheral immune system in the pathogenesis of PD.

## 5. Conclusions

We can conclude that, when compared with healthy subjects, an increased peripheral proinflammatory immune profile was observed in PD patients. Higher NLR, PLR, and ESR values were also found in PD, but only PLR had strong correlations with disease duration and the severity of the disease. These findings could support the role of inflammation in the pathogenesis of PD, but further prospective studies must be carried out on larger groups of participants.

## Figures and Tables

**Table 1 healthcare-10-02578-t001:** Demographics and clinical features of the patients (statistics is given for *t*-test except for *, where a Mann–Whitney U test is reported).

Variables		PD	CG	*p*-Values
n		45	46	-
Age, y		65.91 ± 8.65	62.65 ± 13.39	0.08
Male gender (%)		26 (57.77%)	23 (50%)	1.00
Urban provenience (%)		25 (54.3%)	22 (47.8%)	1.00
Disease duration, y		7.22 ± 5.52	-	-
H&Y stage	1	7 (15.5%)	-	-
	2	13 (28.8%)		
	3	15 (33.3%)		
	4	9 (20.0%)		
	5	1 (2.2%)		
Motor phenotype	Tremor-dominant	12 (26.6%)	-	-
	PIGD	10 22.2%)		
	Indeterminate	23 (51.1%)		
MDS-UPDRS Part III Score		42.2 ± 18.43	-	-
NMSQ		15.33 ± 5.70	-	-
MMSE		27.28 ± 2.15	29.24 ± 0.76	<0.001 *

**Table 2 healthcare-10-02578-t002:** Laboratory data in PD patients and CG (statistics are given for *t*-test except for *, where a Mann–Whitney U test is reported).

Laboratory Data	PD (Mean ± SD)	CG (Mean ± SD)	*p*-Values
n	46	46	-
WBCs (×10^3^/μL)	7.19 ± 1.70	7.32 ±1.69	0.36
Neutrophils (×10^3^/μL)	4.76 ± 1.41	4.58 ± 1.28	0.26
Lymphocytes (×10^3^/μL)	1.89 ± 0.61	2.12 ± 0.72	0.04
Monocytes (×10/^3^μL)	0.38 ± 0.12	0.46 ± 0.15	0.003
Platelets (×10^3^/μL)	257.95 ± 57.26	254.54 ± 64.92	0.89
NLR	2.64 ± 0.94	2.32 ± 0.89	0,04
PLR	168.22 ± 51.48	128.14 ± 40.99	<0.001
ESR	14.60 ± 14.94	8.34 ± 3.45	0.187 *

**Table 3 healthcare-10-02578-t003:** Linear regression models utilized to assess if the different variables predicted the most important clinical parameters in PD patients; the influence of age was also considered as a co-predictor.

Regression Models									
Dependent var.	MDS-UPDRS	MDS-UPDRS	MDS-UPDRS	HY	HY	HY	NMSQ	NMSQ	NMSQ
**Predictor var. 1**	ESR	NLR	PLR	ESR	NLR	PLR	ESR	NLR	PLR
**Predictor var. 2**	Age	Age	Age	Age	Age	Age	Age	Age	Age
**Unstand.β (pred. 1)**	0.143	4.884	0.087	0.014	0.227	0.007	0.084	1.557	0.024
**Unstand.β (pred. 2)**	0.575	0.673	0.48	0.037	0.047	0.031	0.018	0.074	0.021
**N**	44	44	44	44	44	44	44	44	44
**p (pred. 1)**	0.429	0.062	0.118	0.195	0.15	0.027	0.182	0.091	0.228
**p (pred. 2)**	0.07	0.019	0.122	0.046	0.008	0.088	0.868	0.453	0.844
**Adj. R sqr.**	0.085	0.146	0.125	0.143	0.151	0.206	0.009	0.034	0.001

**Table 4 healthcare-10-02578-t004:** Linear regression models utilized to assess the prediction influence between the different clinical and serological parameters, in both the Parkinson and control patients.

Regression Models													
Dependent var.	ESR (PD)	ESR (CG)	NLR (PD)	NLR (CG)	PLR (PD)	PLR (CG)	MMSE (PD)	MMSE (CG)	Disease Duration	Disease Duration	Disease Duration	MDS-UPDRS	NMSQ
**Predictor var. 1**	Age	Age	Age	Age	Age	Age	Age	Age	ESR	NLR	PLR	Onset age	Onset age
**Unstand.β (pred. 1)**	0.655	−0.029	−0.001	−0.018	2.171	−0.007	−0.038	−0.027	0.095	0.767	0.057	−0.037	−0.048
**N**	44	45	44	45	44	45	44	45	44	44	44	44	44
**p (pred. 1)**	0.01	0.451	0.966	0.074	0.008	0.412	0.312	0.001	0.09	0.393	0.001	0.903	0.634
**Adj. R sqr.**	0.124	−0.009	−0.023	0.049	0.133	−0.379	0.001	0.198	0.044	−0.006	0.226	−0.023	−0.018

## Data Availability

The data that support the findings of this study are available on request from the corresponding author.

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
