# Peer review of "The Analysis of Blood Inflammation Markers as Prognostic Factors in Parkinson’s Disease"

_healthcare, 2022, doi:10.3390/healthcare10122578_

Round 1

Reviewer 1 Report

This is an interesting work by Stanca et al., in which the authors analyze candidate inflammatory blood markers in Parkinson’s disease that can be easily quantified during clinical practice. Although the objective is interesting, the approach in novel, and the authors give a good context, I have some issues regarding how the data was interpreted:

1)  First, the authors use T-test for mean comparison, but they do not explain if the data was analyzed to see its distribution. If the distribution is non-normal, non-parametric tests should be used (Mann Whitney U test in this case).

2   2) In Figure 1, authors should the correlation between ESR and motor impairment, showing statistically significant results. It is my feeling that an outlier (top right corner) is driving that relationship. Do the correlation coefficient and significance of the results remain after excluding that one case?

     3) Also, related to the above-comment, as HY is an ordinal variable, may be using “Spearman correlation coefficient” would be more appropriate.

     4) Finally, related to the statistical analyses and interpretation, most of the correlations between blood markers and clinical variables seem to be confounded by age. Although there are not statistical significant differences between PD and healthy controls in terms of age, it is clear from the mean and SD, that controls are overall younger (the distribution is not the same as in PD). Also, considering that disease duration and Motor impairment tend to be higher as patient get older, I think that correlation analyses should be followed with linear regression analyses, while controlling for age. That way, you could really claim that blood inflammatory markers seem to be associated with disease duration/motor impairment. Otherwise, the relationship is pretty flawed.

Some additional comments:

      5) Figures 2, 3 and 4 are really Tables. They should be formatted as tables and adapted to erase irrelevant information (n=46 in each column is unnecessary; ESR within the table in unnecessary).

6)   In Methods, authors explain that PD patients were grouped in 4 subgroups according to their disease duration, in intervals of 5 years. Regarding this:

a.       Why did the authors use 5-year intervals and not another time interval?

b.      They do not provide the number of subjects in each subgroup, nor the mean disease duration within each subgroup (for example, in Table 1).

c.       I feel that this is unnecessary, as they do not compare any variable between disease duration subgroups, right? Authors only use disease duration as a continuous variable in analyses.

Other minor comments:

     7) English writing should be improved.

     8) Authors do not use abbreviations properly. For example, “Central Nervous System” is not abbreviated in lines 58, 60…but use “CNS” in 251. Also, “Parkinson’s disease” is written several times whereas in other occasions “PD” is used instead. Please, use the whole word the first time it appears and add the abbreviation in parenthesis, and use the abbreviation from that on.  

      9) In Table 2 some of the decimals are placed with a comma (,) instead of a dot (.).

      10) Pearson correlation coefficient (r), is internationally indicated lower case letter (see line 118)

Author Response

Q1: First, the authors use T-test for mean comparison, but they do not explain if the data was analyzed to see its distribution. If the distribution is non-normal, non-parametric tests should be used (Mann Whitney U test in this case).

R1: Since we have had roughly more than 30 cases for each group (patients and control), we have indeed overlooked the usefulness of a normality test. Taking the Referee’s suggestion, we have now included a normality test for all the variables described in our statistical analysis. Wherever this test did not support a normal distribution of the data, a Mann Whitney U test has been reported instead of the Student T test.

Q2: In Figure 1, authors should the correlation between ESR and motor impairment, showing statistically significant results. It is my feeling that an outlier (top right corner) is driving that relationship. Do the correlation coefficient and significance of the results remain after excluding that one case?

Q3: Also, related to the above-comment, as HY is an ordinal variable, may be using “Spearman correlation coefficient” would be more appropriate.

Q4: Finally, related to the statistical analyses and interpretation, most of the correlations between blood markers and clinical variables seem to be confounded by age. Although there are not statistically significant differences between PD and healthy controls in terms of age, it is clear from the mean and SD, that controls are overall younger (the distribution is not the same as in PD).

Also, considering that disease duration and Motor impairment tend to be higher as patient get older, I think that correlation analyses should be followed with linear regression analyses, while controlling for age. That way, you could really claim that blood inflammatory markers seem to be associated with disease duration/motor impairment. Otherwise, the relationship is pretty flawed.

R2-R4: We have taken the Referee’s suggestions and tested the Pearson Correlation analysis without the outlier value (ESR =100, MDS-UPDRS = 91). Indeed, the correlation coefficient dropped from 0.416 to 0.234, maintaining a moderate direct correlation. However, it is also true that MDS-UPDRS scale and H&Y stage are categorical variables, and therefore, indeed as suggested by the Referee, a Spearman correlation analysis would have been more indicated compared to a Pearson correlation analysis. Finally, we completely agree with the Reviewer that linear regression modeling would be best to analyses the prediction effect of different factors in influencing other clinical and paraclinical parameters, and wherever possible to test if age could be a strong independent co-predictor in this sense. Therefore, we thank the Reviewer for her/his through suggestions, and our revised manuscript now reports multiple linear regression analysis for dependence evaluation and evaluates the age for a confounder effect. Taking the Reviewer’s analysis, we have completely removed the outlier patient from our data and have reported now 1-tailed statistics in order to explore more precisely differences between data sets.

Q5: Figures 2, 3 and 4 are really Tables. They should be formatted as tables and adapted to erase irrelevant information (n=46 in each column is unnecessary. ESR within the table in unnecessary).

R5: Thank you for your pertinent remarks. When we have generated the new Tables, they are corrected labeled as Tables.

Q6: In Methods, authors explain that PD patients were grouped in 4 subgroups according to their disease duration, in intervals of 5 years. Regarding this:

  1. Why did the authors use 5-year intervals and not another time interval?
  2. They do not provide the number of subjects in each subgroup, nor the mean disease duration within each subgroup (for example, in Table 1).
  3. I feel that this is unnecessary, as they do not compare any variable between disease duration subgroups, right? Authors only use disease duration as a continuous variable in analyses

R6:     a. We used the 5-year interval based on the observation that, on average, after 5 years the severity of the disease changes.

  1. Although we have all the necessary information in the database, we considered that the study would have been difficult to follow with so many variables.
  2. Indeed, since these subgroups were unnecessary in the present study, we did not use them any more in latest version of the manuscript.

Q7: English writing should be improved

R7: We regret there were problems with the English. The paper has been carefully revised.

Q8: Authors do not use abbreviations properly. For example, “Central Nervous System” is not abbreviated in lines 58, 60…but use “CNS” in 251. Also, “Parkinson’s disease” is written several times whereas in other occasions “PD” is used instead. Please, use the whole word the first time it appears and add the abbreviation in parenthesis, and use the abbreviation from that on.

Q9: In Table 2 some of the decimals are placed with a comma (,) instead of a dot

R8-9: We apologize for not using abbreviation properly and putting the comma instead of a dot; we have corrected the text as suggested.

 Q10: Pearson correlation coefficient (r), is internationally indicated lower case letter (see line 118)

R10: We have done the modification suggested by the reviewer.

Thank you very much for your comments that helped us improve this manuscript.

Reviewer 2 Report

In this article the authors aimed to identify novel blood inflammation markers for evaluating the PD prognosis.

The manuscript is well-written and the focus is interesting. However, some points need to be clarified.

As the authors correctly stated, is still unclear whether the chronic inflammation is the cause or the consequence of PD.  The blood-brain barrier integrity is more important to avoid the infiltration of molecules and pathogens

It is significant to note that gut microbiota interact with the brain through systemic chronic inflammation and that the alteration of gut microbiota in PD patients can promote synuclein aggregation and nigrostriastal cell death (please see doi.org/10.3390/app10051828). The intestinal dysfunction it is one of the most common non-motor symptoms in PD patients. That is itself a systemic inflammatory disease that is accompanied by bacterial inflammagens (see doi. 10.3389/fnagi.2019.00210).

Neuroinflammation can be triggered by the presence of specific receptors on the membrane responsible for activation of microglial cells that in turn liberates inflammatory mediators that, with a vicious circle, can cause the synuclein accumulation and aggregation and therefore death of neighbors neurons, the authors should take into account this aspect, i.e. entry, dissemination and infection of virulent bacteria and virus machinery in a systemic manner may be crucial for etiopathogenesis of PD and can affect laboratory data.

On the other hand,

As the inflammation is a very complex process, all these clinical variables, as well as the assumption of probiotics, should be considered during the enrollment of the patients (see https://doi.org/10.3390/cells11162617).

- Table 2: Laboratory data. Blood cell number is incorrectly reported as cells/103/Liters instead of microliters.

- The font of the figure 1 should be increased.

Overall, the intention of this study is intriguing as the identification of non-invasively obtainable liquid markers represents a captivating goal for the diagnosis and prognosis of this devastating pathology

Author Response

Q1: It is significant to note that gut microbiota interact with the brain through systemic chronic inflammation and that the alteration of gut microbiota in PD patients can promote synuclein aggregation and nigrostriastal cell death (please see doi.org/10.3390/app10051828). The intestinal dysfunction it is one of the most common non-motor symptoms in PD patients. That is itself a systemic inflammatory disease that is accompanied by bacterial inflammagens (see doi. 10.3389/fnagi.2019.00210).

Neuroinflammation can be triggered by the presence of specific receptors on the membrane responsible for activation of microglial cells that in turn liberates inflammatory mediators that, with a vicious circle, can cause the synuclein accumulation and aggregation and therefore death of neighbors neurons, the authors should take into account this aspect, i.e. entry, dissemination and infection of virulent bacteria and virus machinery in a systemic manner may be crucial for etiopathogenesis of PD and can affect laboratory data.

On the other hand, As the inflammation is a very complex process, all these clinical variables, as well as the assumption of probiotics, should be considered during the enrollment of the patients (see https://doi.org/10.3390/cells11162617).

R1: Thank you for this suggestion. It would be interesting to explore this aspect in the future.  Our study is a retrospective one and none of the patients included in this study were administrated probiotics. It is known that Parkinson's disease (PD) is a neurodegenerative disease, which, according to the recent studies, is associated with systemic inflammation, especially with intestinal microbiota inflammation. Bacteria colonize in large numbers the human intestine with a protective role, against pathogens, so that an imbalance of the commensal flora by destroying and disrupting the permeability of the intestinal barrier can cause pathogens and pro-inflammatory substances to enter the digestive tract. Also, the intestinal microbiota can determine the initiation of α-synuclein aggregation in the enteric system. In this sense, there is a wide variety of commercial probiotics, but their effectiveness depends on whether these products contain one or more strains. More studies are needed to determine if treatment with probiotics or antibiotics in patients with PD would reduce systemic inflammation and alpha synuclein aggregation. Given the fact that our study is a retrospective one that did not have as its topic the intestinal microbiota of the patients with PD, no references were made to this subject, considering that the involvement of the intestinal microbiota in the pathology and evolution of PD requires exhaustive studies, both prospective and retrospective.

Neuroinflammation is associated with many neurodegenerative diseases, including Parkinson's disease. Both microglia and astrocytes are important regulators of inflammatory responses in the central nervous system.

The release of aggregated pathogenic proteins, such as α-synuclein, into the extracellular space leads to changes in microglia and astrocytes, which determines an increase of pro-inflammatory factors and a decrease in the phagocytic effect. This causes impairment of synaptic function, blood-brain barrier, metabolic function and extracellular ions. In PD, α-synuclein overexpression leads to proinflammatory microglia that releases TNF-α, NO and IL-1β that will modulate the neuroinflammatory process in this disease.

However, the role of the neuroprotective microglial phenotype and the pathological mechanisms underlying neurodegenerative diseases are not fully understood. From the present studies there are certain markers that may indicate neuroinflammation, such as Arginase 1 (converts L-arginine into ornithine), Ym1 (a secretory protein that links heparin/heparan sulfate), CD163 (protein that binds hemoglobin-haptoglobin complex), Dectin-1 (lectin receptor that recognizes β-glucans and stimulates phagocytosis).

For the future, in a prospective study, it would be of real interest to track these markers, given the fact that our study was retrospective, and the enrolled patients did not present infections of the digestive tract or systemic infections

Q2: Table 2: Laboratory data. Blood cell number is incorrectly reported as cells/10 /Liters instead of microliters.

R2: Revised accordingly

Q3: The font of the figure 1 should be increased.

R3: Given the fact that we did another statistical analysis, we replaced the figure and the change is no longer necessary.

 Comment: Overall, the intention of this study is intriguing as the identification of noninvasively obtainable liquid markers represents a captivating goal for the diagnosis and prognosis of this devastating pathology.

Appropriate changes regarding the English language were made.

Thank you very much, we appreciate the reviewer’s feedback.

Reviewer 3 Report

Healthcare-2073014

Title: The analysis of blood inflammation markers as prognostic factors in Parkinson’s disease

The work consists of the evaluation of several biochemical parameters and calculation of ratios in healthy population and compare them with patients with Parkinson's disease (PD).

The authors found differences between both populations in two ratios (PLR and ESR).

The paper is well written, easy to read and without serious methodological errors.

I would like to make the following considerations:

-      In a research paper on Parkinson's disease, the importance of alpha-synuclein and its role in inflammation should be mentioned.

-      In vivo (line 67) is a Latin term, so it should be written in italics.

-      Blood samples are taken and analyzed from patients, so the authorization of the corresponding bioethics committee must be explicitly mentioned, including the corresponding number.

Author Response

Q1: In a research paper on Parkinson's disease, the importance of alpha-synuclein and its role in inflammation should be mentioned.

R1: Thank you for your nice reminder. The importance and the role of alpha-synuclein is well known in the pathogenesis of Parkinson's disease. We know that the two main pathological hallmarks of this disease are the loss of dopaminergic neurons and the alpha-synuclein accumulations. There is a lot to say about the pathogenesis of Parkinson's disease which is still not clear, many studies carried out, many hypotheses and many variables to consider. Although we mentioned about alpha-synuclein in the section “DISCUTIONS” we updated the text with some information about the role of alpha-synuclein in inflammation in the section of “INTRODUCTION”

Q2: In vivo (line 67) is a Latin term, so it should be written in italics.

R2: We thank the Referee and we made the suggested change.

Q3: Blood samples are taken and analyzed from patients, so the authorization of the corresponding bioethics committee must be explicitly mentioned, including the corresponding number.

R3: Thank you for pointing this out. We mentioned this at the end of the article at” Institutional Review Board Statement” and ”Informed Consent Statement” but now we clearly specified in the text of the article.

Thank you for your time again, we look forward to hearing from you in due time regarding our submission and to respond to any further questions and comments you may have.

Round 2

Reviewer 1 Report

I would like to congratulate the authors for their work and throughness in improving this manuscript. I only have few minor comments and suggestions to finalize the review:

1. In the abstract, the authors say "strong" correlation between PRL and disease stage and disease duration. I feel that they meant "significant" correlation, as the estimated coefficient do not show a strong correlation, rather it is a weak correlation, but is significant. 

2. In methods, the author say that they perfomed "multiple regression analyses". Please, specify that this were multivariable regression analyses using "age" as the main confounder (or covariable or that the analyses were adjusted for age). They mention this in the results, but it should also appear in methods.

3. Table 1. May be it should be written below the table that categorical variables are indicated in the table as number (percentage) whereas continuous variables are indicated as mean +/- SD. 

4. P values 0.000 shoud be writeen as p < 0.001. This happens in Table 2 and in section 3.1.3. Please, review that there are no more p=0.000 in the text, and if there are, change accordingly.

5. Please, verify that in the final version of the manuscript, in the last sentence of the "Statistical analysis" that there is no dot missing. 

"...without normal distributionThrough the analysis..."

I don't know if the tracking system failed here, but a dot is missing or may be "through the analysis" should not be there... revise the sentence, please.

Author Response

I would like to congratulate the authors for their work and throughness in improving this manuscript. I only have few minor comments and suggestions to finalize the review:

Comments 1. In the abstract, the authors say "strong" correlation between PRL and disease stage and disease duration. I feel that they meant "significant" correlation, as the estimated coefficient do not show a strong correlation, rather it is a weak correlation, but is significant. 

R1) Thank you for pointing this out. We agree with this comment, therefore we made the change.

Comment 2. In methods, the author say that they perfomed "multiple regression analyses". Please, specify that this were multivariable regression analyses using "age" as the main confounder (or covariable or that the analyses were adjusted for age). They mention this in the results, but it should also appear in methods.

R2) We have, accordingly, revised and mention the age as a confounder in the methods.

Comment 3. Table 1. May be it should be written below the table that categorical variables are indicated in the table as number (percentage) whereas continuous variables are indicated as mean +/- SD. 

R3) Thank you for this suggestion. We agree with this and have incorporated your suggestion throughout the manuscript.

Comment 4. P values 0.000 shoud be writeen as p < 0.001. This happens in Table 2 and in section 3.1.3. Please, review that there are no more p=0.000 in the text, and if there are, change accordingly.

R4) Revised accordingly  

Comment 5. Please, verify that in the final version of the manuscript, in the last sentence of the "Statistical analysis" that there is no dot missing. 

"...without normal distributionThrough the analysis..."

I don't know if the tracking system failed here, but a dot is missing or may be "through the analysis" should not be there... revise the sentence, please.

R5) We apologize for this error, and we have corrected the text as suggested.

Thank you very much and we appreciate very much your feedback.